# Iron Status is Associated with Mood, Cognition, and Functional Ability in Older Adults: A Cross-Sectional Study

**DOI:** 10.3390/nu12113594

**Published:** 2020-11-23

**Authors:** Carlos Portugal-Nunes, Teresa Costa Castanho, Liliana Amorim, Pedro Silva Moreira, José Mariz, Fernanda Marques, Nuno Sousa, Nadine Correia Santos, Joana Almeida Palha

**Affiliations:** 1Life and Health Sciences Research Institute (ICVS), School of Medicine, University of Minho, 4710-057 Braga, Portugal; carlosportugalnunes@gmail.com (C.P.-N.); maat1087@gmail.com (T.C.C.); id5225@alunos.uminho.pt (L.A.); pedromsmoreira@gmail.com (P.S.M.); briote@gmail.com (J.M.); fmarques@med.uminho.pt (F.M.); njcsousa@med.uminho.pt (N.S.); nsantos@med.uminho.pt (N.C.S.); 2ICVS/3B′s, PT Government Associate Laboratory, 4710-057 Braga/Guimarães, Portugal; 3Clinical Academic Center—Braga, 4710-243 Braga, Portugal; 4Associação Centro de Medicina P5 (ACMP5), 4710-057 Braga, Portugal; 5Emergency Department, Intermediate Care Unit (EDIMCU), Hospital de Braga, 4710-243 Braga, Portugal

**Keywords:** iron deficiency, mood, cognition, physical functional ability, aging

## Abstract

Several conditions are risk factors for iron deficiency (ID), some of which are highly prevalent in older individuals. Despite the amount of evidence pointing for a role of ID in cognition, mood and physical functional ability, the research addressing these associations in older individuals is still scarce. In the present study, 162 older community-dwelling individuals (29.53% classified as ID) were enrolled in a cross-sectional analysis and characterized regarding cognition, mood, functional ability, general nutritional intake and iron status. Assessment of iron status was performed using several blood biomarkers. Storage and erythropoiesis dimensions were positively associated with memory, along with an interaction (moderator effect) between iron storage and nutritional status. A more depressed mood was negatively associated with (iron) transport, transport saturation and erythropoiesis dimensions, and functional tiredness was positively associated with the erythropoiesis dimension. These observations indicate that lower iron status is associated with depressive mood, functional tiredness and poorer memory ability, with the latter moderated by nutritional status. These findings suggest that using iron as a continuous variable may be useful in finding associations with iron homeostasis, eventually missed when iron levels are considered within the usual classification groups.

## 1. Introduction

Imbalance in iron homeostasis, both excess and deficiency, are deleterious to human health and have been associated with medical conditions; these include neurodegenerative disorders (i.e., Parkinson and Alzheimer diseases), type II diabetes, and anemia [1,2,3]. Paradoxically, although iron is one of the most abundant elements on the planet, iron deficiency (ID) is the most common nutritional deficiency [4]. The World Health Organization estimates that more individuals have ID anemia than any other health problem [5]. The rise of life expectancy in the last century has led to an increase of the older population worldwide [6]. This current demographic and societal phenomenon will result in an increasing number of older individuals with various age-associated health problems and pathologies [7,8]. Furthermore, older individuals are also the largest consumers of prescribed drugs [9]. On this, of relevant note, several morbidities (e.g., gastric and duodenal ulcers, adenomatous polyps and erosive gastritis) and therapeutic drugs (e.g., antacids, H_2_ antagonists, proton pump inhibitors, aspirin or nonsteroidal anti-inflammatory drug), that are prevalent in older individuals, are also possible causes of ID [10].

Although a matter of concern across all age groups; infants, adolescents, women of childbearing age or pregnant, and older individuals, are particularly susceptible to ID [11]. In older adults, ID may be caused by insufficient dietary iron intake, malabsorption or blood losses from the gastrointestinal tract [12]. In adults, data from epidemiological and interventional studies indicate that ID can result or be associated with a wide range of adverse effects, including: fatigue [13,14], reduced work performance, diminished exercise capacity [15], impaired thermoregulation [16], immune dysfunction [17], and neurocognitive impairment. The potential negative impact of ID in cognition at older ages may come from cerebral hypoxia, poor myelin integrity or insufficient neurotransmitter synthesis [12].

Taken together, aging and ID can negatively impact on health and wellbeing, including in cognitive and physical ability capabilities. Notably, extremely little focused research regarding ID has been conducted in this population stratum. A systematic review and meta-analysis of epidemiological longitudinal studies [18] found an increased risk of incident dementia in anemic individuals; however, the type of anemia was not addressed. Although anemia is used as an indicator of ID and the terms anemia, ID and ID anemia are used interchangeably, it should be noticed that anemia can also be caused by vitamin B12 deficiency, which is a well-known cause of dementia [19,20]. Furthermore, physical functional ability in the elder has been associated with anemia [21,22,23,24]. Still, to the best of our knowledge no study has clearly addressed the potential implications of ID in older individuals, which here we performed by exploring the association between iron status, cognitive ability and physical functional performance in a cohort of community-dwelling middle-aged/older individuals, without dementia and/or neuropathology.

## 2. Materials and Methods

### 2.1. Subjects

Since the first aim of this work is to explore differences in cognition, mood and physical functional ability between individuals with ID and iron sufficiency, sample size was calculated using GPower (v. 3.1.9.7 program written by Franz Faul, Universität Kiel, Germany) for the difference between two independent means considering a medium effect size (d = 0.5), power = 0.8, α = 0.05 and allocation ratio iron sufficiency group/ID group = 0.3. This estimation indicated a sample size of 142 individuals (iron sufficiency group = 109 and ID group = 33).

A convenience sample of 303 individuals was contacted and invited to participate from primary health care centers (Braga and Guimarães/Vizela) and internal medicine outpatient care (Hospital de Braga), until the calculated sample size was reached. We further included 20 additional participants to account for exclusions and drop-outs. Participants were community-dwelling individuals aged 55 years or older, males and females, with a general good health status, integrated in the community and with independency/autonomy to perform the activities of the daily living. Initial exclusion criteria included incapacity and/or inability to attend the assessment sessions, cognitive impairment, dementia diagnosis and/or inability to understand informed consent, disorders of the central nervous system and/or overt thyroid pathology. The assessment of exclusion criteria was initially based on self-report; the presence or absence of medical conditions was next confirmed from medical records. As shown in Figure 1, *n* = 162 participants accepted to participate in the study. From those, 11 subjects were further excluded due to: malnutrition (*n* = 2, mini-nutritional assessment (MNA) score < 17), chronic kidney disease (*n* = 1; creatinine > 2.5 mg/dL), high inflammatory status (*n* = 7; high sensitivity C reactive protein (hsCRP) >10 mg/L) and iron overload (*n* = 1; transferrin (TF) saturation > 55%). Other exclusion criterium was severe anemia (hemoglobin < 9 mg/dL), but no participant presented such criterium. After exclusion, the study sample comprised 149 individuals [females, *n* = 81 (54.4%); males, *n* = 68 (45.6%)]. In some measures, we were not able to collect data from participants, due to refusal of participants or due to contraindications, such as pacemaker in bioelectrical impedance analysis (BIA). Even with missing data for some measures we decided to use all available data from each participant. The sample size for each variable is presented as footnote in the tables.

The cohort was established in accordance with the principles expressed in the Declaration of Helsinki and the work approved by the national ethical committee (Comissão Nacional de Protecção de Dados) (approval n.° 352/2011–07/02/2011) and by local ethics review boards (approval n.° 10/CES–07/05/2010). The goals and nature of the tests were explained to potential participants and all volunteers provided informed consent. Socio-demographic characteristics and clinical measures were self-reported and confirmed from medical records.

### 2.2. Laboratory Analyses

Blood samples were collected by venipuncture before the cognitive and functional ability assessments and immediately sent to the Pathology Laboratory at the Hospital de Braga for analysis. Blood cells count and hemogram were performed using certified standardized methods and comprised red blood cells count (RBC; 10^12^/L), hemoglobin (mg/dL), hematocrit (%), mean corpuscular volume (MCV; fL), mean corpuscular hemoglobin (MCH; pg), mean corpuscular hemoglobin concentration (MCHC; g/dL) and red cell distribution width (RDW; %). Serum iron (Fe; µg/dL) and total iron binding capacity (TIBC; µg/dL) were determined by a colorimetric method using Dimension Vista System Flex reagent cartridge (Siemens, Frimley, Camberly, UK). High sensitivity C reactive protein (hsCRP; mg/dL), transferrin (TF; mg/dL), ferritin (FT; ng/mL) and serum concentration of soluble transferrin receptors (sTFR; mg/dL) were measured by chemiluminescent immunoassays. Dimension Vista System Flex reagent cartridge (Siemens, Frimley, Camberly, UK) was used to measure TF, FT and sTFR; the BN* II and BN ProSpec System (Siemens, Frimley, Camberly, UK) was used to measure hsCRP. All determinations were performed following the manufacturers’ instructions. Detection limits for hsCRP, Fe, TIBC, TF, FT and sTFR were 0.175 mg/dL, 5 µg/dL, 8 µg/dL, 8.75 mg/dL, 0.5 ng/mL and 0.017 mg/L, respectively. TF sat. (%) was calculated as a percentage of serum total iron divided by TIBC. The sTFR-logFT index was calculated by sTFR divided by the logarithm of FT. Body iron was calculated using the Cook algorithm [25] as follows: body iron (mg/kg) = −[log(sTFR*1000/FT) − 2.8229/0.1207]. ID was defined as low serum FT level (FT < 15 ng/mL) or as the presence of two biomarkers indicating ID as described elsewhere [26,27,28] (MCV < 80 fL, MCHC < 32 g/dL, RDW > 14%, Fe < 71 µg/dL, TF sat. < 20%, TIBC ≥ 360 µg/dL, sTFR > 1.76 mg/L and sTFr-logFT index > 1.5).

### 2.3. Neurocognitive and Physical Functional Assessment

Tests were selected to provide cognitive profiles (general cognitive status, and executive and memory functions), as previously reported [29,30]. The cognitive/psychological characterization was performed by a team of trained psychologists following the instructions provided in a standard operating procedures manual. The following cognitive measures were used: global cognitive status was assessed with the mini-mental state examination (MMSE) [31]; short-term verbal memory with the digit span (DS) forward test (DS forward; subtest of the Wechsler adult intelligence test—WAIS III), verbal working memory with the DS backward test (DS backward; subtest of the Wechsler adult intelligence test WAIS III) and DS total score (DS total; calculated by the summation of DS forward and DS backward) [32]; multiple trial verbal learning and memory with the selective reminding test [SRT—List A; parameters: consistent long term retrieval (CLTR), long term storage (LTS), delayed recall (DR) and intrusions] [33], and the Consortium to Establish a Registry for Alzheimer′s disease-word list test [CERAD, parameters: total hits and DR hits] [34]; response inhibition/cognitive flexibility with the Stroop color and word test [Stroop, parameters: words (W), colors (C) and words/colors (W&C)] [32]. The geriatric depression scale (GDS, long-version) [35] was used for depressive mood evaluation; higher values represent a more depressive mood. The physical functional ability to perform physical activities of daily living (PADL) was assessed using the questionnaire of physical functional ability (QoFA) [36]. This instrument is composed of three subscales assessing tiredness, [(i) mobility tiredness, (ii) lower limb tiredness, and (iii) upper limb tiredness] and of two subscales assessing dependency [(iv) mobility help and (v) PADL help]. Higher values in each subscale represent higher functionality. The instrument was applied by a registered dietitian/nutritionist after a brief explanation of its structure and aims.

### 2.4. Nutritional Status and Anthropometric Characterization

The full version of the MNA was used to identify malnourishment (MNA score < 17) or those at risk of malnutrition (MNA score from 17 to 23.5) [37]. MNA is a validated questionnaire designed to provide a single and rapid measure of nutritional status in older individuals. It is composed by anthropometric measures, questions related to medication, mobility, autonomy of feeding, number of meals, food and fluid intake and self-perception on health and nutrition [38]. With respect to food intake, there are three questions (at least one serving of dairy products (milk, cheese, yoghurt) per day; two or more servings of legumes or eggs per week and meat, fish or poultry every day). Questions were applied in a face-to-face interview. Data on body mass index (BMI), mid arm circumference and calf circumference were obtained during the anthropometric characterization. Weight and relative body fat mass (%BF) were measured with the participants wearing light wear using a Tanita^®^ BF 350 Body Composition Analyzer (Tanita Corporation, Tokyo, Japan), which uses the foot-to-foot BIA to estimate %BF. The output variables were calculated according to the manufacturer’s embedded software. Height was measured without shoes using a stand-alone stadiometer Seca^®^ 217 (Seca GmBH & Co Kg, Hamburg, Germany).

### 2.5. Statistical Analysis

Characteristics of the participants are presented as mean and standard deviation (mean; SD) for normal distributed variables and as median and interquartile range (median; IQR) for variables with a non-normal distribution. To evaluate normal distribution of the variables, skewness and kurtosis values were calculated and the approximate normal distribution was defined for variables with absolute values of skewness below 3 and of kurtosis below 8 [39]. Log transformations were performed to normalize the distribution of skewly distributed variables (hsCRP, FT; sTFR and sTFR-logFT index). Independent samples t-test (for variables with normal distribution) and Mann–Whitney U test (for variables with non-normal distribution) were performed to analyze the differences in socio-demographic, anthropometric, psychological, cognitive, physical functional ability and hematological variables between individuals with or without ID. Differences in categorical variables were assessed using Chi-squared test. When assumptions for Chi-square tests on contingency tables were violated, the two tailed significance level of Fisher exact test was used. All variables were converted into z-scores so to be expressed in the same scale.

Principal component analysis (PCA) was conducted to reduce the number of variables with a minimum loss of information and, therefore, reducing the number of comparisons. For these analyses, log-transformation of non-normal distributed variables transformed into z-scores (FT, sTFR and sTFr-logFT index) were used. In some cases, z-scores of variables were inverted to make higher values represent higher iron status (RDW, TF, TIBC, sTFR and sTFr-logFT index). New component scores were obtained (using the regression method) and were used in subsequent analyses. The reliability of each component was analyzed using Cronbach’s alpha. Components were considered reliable when Cronbach’s alpha was higher than 0.6 [40]. Variables not included in PCA were analyzed independently.

ANCOVA was used to test differences in dependent variables (psychological, cognitive and physical functional ability) between individuals with or without ID, controlling for the principal confounding factors (age, education, gender and hsCRP for psychological variables; the previous plus GDS score for cognitive variables and dimensions; and age, BMI and hsCRP for physical functional ability dimensions). Education was converted to a dummy variable (school years; <4 = 0; ≥4 = 1) due to the high number of individuals with four years of formal education.

Binary logistic regression analysis was used to test whether nutritional status (MNA) and body composition (BMI and %BF) were significant predictors of ID when controlled for potential confounding factors (age, gender and hsCRP).

Hierarchical regression analysis was performed to test different hematological dimensions (that resulted from the PCA) as predictors of the previously mentioned dependent variables controlling for the principal, abovementioned, confounding factors.

A moderation analysis was performed to test the significance of the interactions of MNA and hematological dimensions, adjusted for the confounding factors above mentioned, using PROCESS v3.5 for IBM SPSS Statistics [41]. A statistically significant interaction indicates that the moderator variable (MNA) changes the strength or trend (positive/negative) of the association between the dependent variable and the independent variable (hematological dimensions). To allow the visualization of the moderation effect, pick-a-point plots were obtained using the syntax provided on PROCEESS output at mean and mean ± 1 SD. The Johnson–Neyman technique was used to identify the regions of significance for the moderator.

Statistical analysis was conducted using the SPSS package v25 (IBM SPSS Statistics, IBM Corp., Armonk, NY, USA) and statistical significance was defined at *p* < 0.05 level.

## 3. Results

### 3.1. Sample Characterization

Characterization on iron status is presented in Table 1. From the 149 participants, *n* = 44 individuals (29.53% |ID = 36; ID anemia = 8) were classified as ID and *n* = 105 individuals (70.47% | anemic = 15; normal = 90) as iron sufficient. No participant presented severe anemia (Hb < 9mg/dL). Height was significantly higher in the iron sufficient which can be attributed to the differences in gender distribution since more male individuals were in this group. Iron sufficient participants presented a statistically significant higher score in the MNA; yet, no significant differences in the distribution of risk of malnutrition was observed between iron sufficient and ID participants. It is important to highlight that in this work we excluded malnourished individuals and the mean value of the MNA score in both groups is high. While within the normal range, hsCRP was significantly higher in the ID group. 

### 3.2. Mood, Cognitive and Functional Characterization

Two cognitive dimensions were obtained from the PCA, termed: (i) executive dimension and (ii) memory dimension. The executive dimension (Cronbach’s alpha: 0.880) was composed of the Stroop (W, C and W&C) and DS (backward and total) parameters. The variables DS forward and MMSE total score were included in the initial analysis; however, since they provided low communalities, were later removed from the final model. The memory dimension (Cronbach’s alpha: 0.933) was composed of the SRT (CLTR, LTS and DR) and the CERAD (total hits and DR) parameters. Descriptive statistics for psychological and cognitive variables for iron sufficient and iron deficient individuals are presented in Table A1. As show in Figure 2, GDS presented a higher mean value in the ID group. No other significant differences between groups were observed.

Similarly, two dimensions were obtained from the subscales of the QoFA: (i) functional-T dimension (functional tiredness, Cronbach’s alpha: 0.763), composed by mobility tiredness, lower limb tiredness and upper limb tiredness subscales; and (ii) functional-H dimension (functional help, Cronbach’s alpha: 0.690), composed by mobility help and PADL help. Like the variables of origin, higher scores represent higher functionality, so higher scores on functional-T represent less tiredness and higher scores on functional-H represent less need of help. Data from physical functional ability (Table A2) showed no statistically significant differences between iron sufficient and deficient participants.

After controlling for principal confounding factors, there were no differences between groups of iron status for mood, cognition and physical functionality (Table A3).

### 3.3. Hematological Characterization

Similar to previously described by Murray–Kolb et al. [26], dimensions of hematologic variables were obtained (here using PCA). Specifically, five components were obtained using the hematologic variables, four of which were similar to the factors described by Murray–Kolb et al. [26]: (i) storage, (ii) transport, (iii) red cells characteristics (red cells C) and (iv) erythropoiesis. A new component was obtained from the remaining biomarkers: (v) transport saturation (transport S). Storage dimension (storage of iron; Cronbach’s alpha: 0.925) was composed by FT, body iron, sTFR and sTFr-logFT index. Transport dimension (iron transport in blood stream; Cronbach’s alpha: 0.856) included Fe and TF sat. Transport S dimension (saturation of iron carrying capacity in the blood stream; Cronbach’s alpha: 0.979) was obtained by the reduction of TF and TIBC. Red cells C dimension (composition, dimension and variability of red cells; Cronbach’s alpha: 0.822) constructed with MCV, MCH, MCHC and RDW. Erythropoiesis dimension (number and relative volume of red cells and hemoglobin sufficiency; Cronbach’s alpha: 0.967) composed by RBC, hemoglobin and hematocrit. Higher values in these dimensions represent higher iron status. As expected, significant differences were observed for almost all hematologic variables and dimensions between groups of iron status (Table A4), with the iron sufficient group presenting higher values.

### 3.4. Nutritional Status is a Predictor of ID

To test whether nutritional status (MNA) and/or body composition (BMI and %BF) were predictors of ID, logistic regression models were conducted (Table A5). For all models, gender and hsCRP (log) were significant predictors of ID. Indicating that the odds of ID presence are lower (OR = 0.197 to 0.398) in males than in females, and for each fold increase in hsCRP an increase in the odds of the presence of ID is observed (OR = 7.834 to 9.625). The Wald criterion demonstrated that nutritional status, measured by the MNA total score, made a significant contribution to prediction of ID (Wald_(1)_ = 5.389; *p* = 0.020). For each unit increase (1 point in the MNA total score) in nutritional status the odds of ID presence decreased 1.211 times (OR = 0.826). Body composition variables did not contribute significantly to the prediction of ID.

### 3.5. Iron Hematological Dimensions Predict Memory, Mood and Functional Ability

Results from hierarchical regression models are presented in Table 2. All final models significantly explained the dependent variables. Memory and GDS scores were significantly predicted by the hematological dimensions considered. Specifically, memory by storage and erythropoiesis dimensions, and GDS by transport, transport S and erythropoiesis. For the functional dimensions, only erythropoiesis predicted functional-T.

Results from the moderation analysis testing the effect of nutritional status (MNA) on the association between hematological dimensions and cognition, mood and functional ability are presented in Table A6. The only significant MNA moderation effect was observed in the association between storage and memory. Johnson–Neyman significance regions indicate that for MNA values above 26 the association is positive and statistically significant, becoming stronger with the increase in the MNA score (Figure 3).

## 4. Discussion

The participants enrolled in this study were cognitively healthy middle-age/older community dwellers and were classified as ID (or not) using a comprehensive panel of iron biomarkers. Iron research usually distinguishes between iron sufficient and deficient in accordance with established cut-offs. Using this approach, the present study did not find associations between iron status, cognition, mood and physical ability. In other populations (namely children, adolescents and women of childbearing age), ID has been associated with impairments on cognition, mood and physical performance [42,43,44,45]. Here, no differences were seen between the iron deficient and sufficient groups, probably because ID was not sufficiently severe. From the literature it is clear that the focus of low iron status research and cognition has been based on the categorical classification of ID, particularly in children, anemic individuals [46,47,48,49], and animal models [50]. Here, the number of subjects in each condition (ID = 36; ID anemia = 8; anemic = 15; normal = 90) prevented us from conducting an analysis regarding anemia or ID anemia. Interestingly, when iron status is considered as a continuum, and after controlling for the main confounding factors, iron status is associated with the memory dimension, GDS and with functional tiredness. These findings suggest that, from the point of view of cognitive function, mood and physical ability, classifying individuals as ID may not necessarily constitute the best strategy [51].

We next approached iron homeostasis considering function dimensions (hematologic variables) [26]. The erythropoiesis dimension was associated with memory but not with the executive dimension or MMSE. This is consistent with the findings by Shah et al. [52] who reported lower memory ability in individuals with lower levels of hemoglobin. Similarly, the iron storage dimension was a predictor of memory. Interestingly, the iron storage dimension was also a predictor of memory. ID is recognized to be deleterious for learning and memory, particularly when occurring in critical time windows of brain development [50]. These effects, which may remain into adulthood [53,54], have been associated with abnormal hippocampal structure and plasticity [50]. No other associations were observed between the remaining hematological variables and memory and executive function. Interestingly, others have described an association between ID and MMSE in “healthy” older adults [28]. In other population groups, namely young women, iron status has been associated with executive function and cognition; however, in some cases, these associations were found between the time to complete a task and not the result of the task (as we did here) [26,55]. During the aging process, a certain degree of brain iron accumulation is observed and may be deleterious [56]. Insights from experiments on animal models of ID provide clues that possibly explain this finding; particularly in brain areas related to memory such as the hippocampus [57]. ID has been shown, in rats, to influence resting energy status, neurotransmission and myelination [58]. The precise molecular mechanisms mediating its effects, particularly during aging, remain to be identified and are likely to involve molecules such as hippocampal brain-derived neurotrophic factor [59].

Erythropoiesis, transport S and transport dimensions were significant predictors of GDS score, which is concordant with previous studies examining the association of anemia and depressive mood. In the InChianti study [60] a higher prevalence of anemia was observed among the individuals displaying depressive symptoms. In the same line, the results of the English Longitudinal Study of Ageing [61] showed that, at baseline, anemia was associated with depressive mood. More recently Stewart et al. [62] found anemia to be associated with depressive mood; although, it was reflecting primarily the anemia of chronic disease. This result should be interpreted with caution, since inflammation is a component of anemia of chronic disease and is associated with depression per se [63].

Finally, we observed that erythropoiesis dimension was associated with the tiredness dimension of physical functional ability. A possible explanation for the fatigability observed in individuals with lower hematological dimension scores is the lower oxygenation of the muscles [22,51]. Literature in the study of ID and physical ability in older subjects is surprisingly scarce, contrary to the large amount of literature examining the same in anemia [21,22,23,24]. In iron-depleted nonanemic women, iron supplementation was associated with a significant improvement in muscle fatigability [64]. Furthermore, ID, particularly ID anemia, was associated with a compromised aerobic and endurance capacity in a large number of studies from animal models to humans and it was hypothesized that, in addition to deficient oxygen delivery, tissue ID may also play a role through reduced cellular oxidative capacity [15].

It has been proposed that ID could be a surrogate marker for malnutrition [65,66]. Although the MNA test was designed to detect protein-energy malnutrition and not micronutrient deficiency [67], some authors [38,68] found significantly lower intakes of iron in individuals with low MNA scores. In addition, iron replacement treatment was shown to have a positive impact on nutritional status of older patients with ID and ID anemia [69]. Even though in theory, MNA scores can be associated with iron status, to our knowledge, no studies had directly investigated this relationship. In our population we observed that ID can be significantly predicted by MNA. Importantly, nutritional status has been for a long time described to be associated with cognition [70], mood [71] and physical performance [72].

When addressing the moderation effect of nutritional status, we observed that the only interaction with significant predictive value was MNA and storage on memory. This may indicate that, for individuals with higher stores of iron, if in a good nutritional status, memory function will be at a potential maximum, indicating that higher levels of insufficiency are necessary to impact on individuals with a better nutritional status. This is an interesting novel finding that deserves further investigation.

Of notice, even within levels that exclude inflammatory states, hsCRP values differed between iron sufficiency and ID cases, being higher in the latter. As expected, the ID group presented significantly different values in the red cells indices, iron biomarkers and hematological dimensions, with exception for RBC. The occurrence of hypoferremia during inflammation has been recognized for more than a half of century [73]. Furthermore, it is known that inflammatory hallmarks are present in the etiology of anemia of chronic disease and that this relation is mediated by hepcidin, which is overexpressed in inflammatory states and negatively regulates iron availability [74,75]. Here, we cannot conclude whether the classification of ID is absolute or functional, given the higher levels of hsCRP in the ID group [76].

In an aging society, efforts to understand and identify factors that improve or maintain health and wellbeing of agers are paramount. Cognition, mood and physical functioning are particularly relevant to maintain independence. To the best of our knowledge this is the first population-based observational study that examines the associations of iron status with cognitive function, psychological morbidity and functional ability in middle-age/older community dwellers. As for all the observational studies, no causality relationship can be established. In addition, the use of a convenience sample recruitment strategy may be associated with selection bias; still, this does not hamper the associations observed.

In summary, the viewpoint that ID does not have consequences until the development of anemia is here challenged. Considering iron status as a continuum may provide relevant information to assess the consequences of low iron levels. This is accordance with the observations by Yavuz et al. [28], who showed that ID has negative consequences on cognitive function independently of the presence of anemia. Furthermore, ID seems to be deleterious even in the absence of erythropoietic effects [77]. In conclusion, low iron status in older individuals is associated with higher depressive mood, higher tiredness, and lower memory, which seem to be modulated by nutritional status. Further research is needed to replicate and confirm the present findings, including studies with longitudinal or interventional designs.

## Figures and Tables

**Figure 1 nutrients-12-03594-f001:**
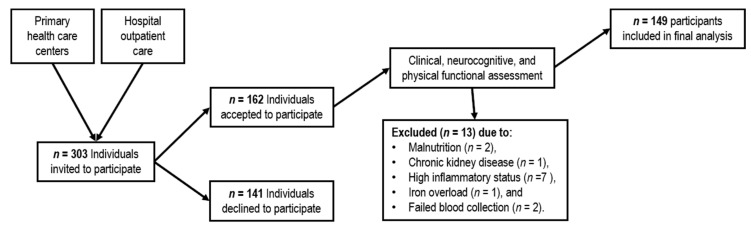
Schematic representation of recruitment and inclusion of participants in the study.

**Figure 2 nutrients-12-03594-f002:**
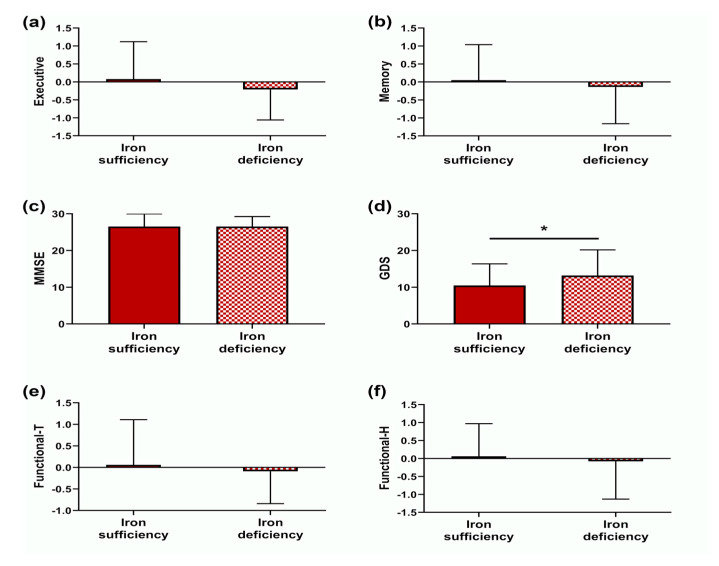
Cognition, mood and physical functional ability comparison between iron sufficient and iron deficient participants. (**a**) Executive dimension (z-score); (**b**) Memory dimension (z-score); (**c**) Mini-mental state examination (MMSE) score; (**d**) Geriatric depression scale (GDS) score; (**e**) Functional tiredness dimension (z-score); (**f**) Functional help dimension (z-score). * *p* < 0.05.

**Figure 3 nutrients-12-03594-f003:**
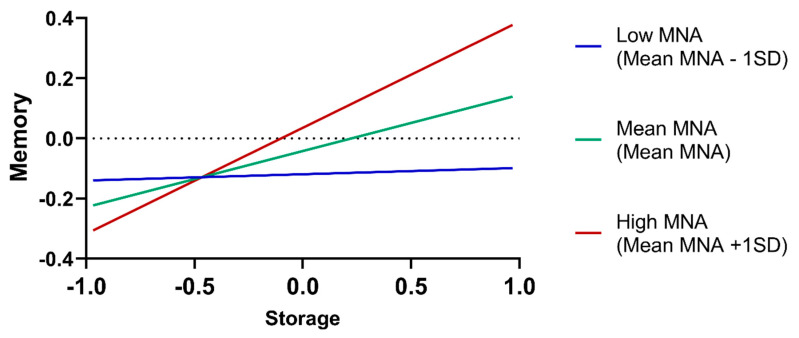
Adjusted association of storage dimension with memory according to nutritional status (mini nutritional assessment, MNA).

**Table 1 nutrients-12-03594-t001:** Characteristics of participants and differences by group of iron status.

Variables	Iron Sufficiency	Iron Deficiency	
**Socio-Demographic (Mean; SD)**			**t_(df)_; *p*; Cohen’s d**
Age (years)	66; 8	66; 7	0.197_(147)_; 0.844; 0.036
Education (school years)	5; 4	5; 4	−0.043_(147)_; 0.966; 0.008
**Anthropometric (mean; SD)**			**t_(df)_; *p*; Cohen’s d**
Weight (kg) ^a^	74; 2	71; 2	1.426_(138)_; 0.156; 0.267
Height (m) ^a^	1.60; 0.08	1.56; 0.08	2.218_(138)_; 0.028; 0.415
BMI (kg/m2) ^a^	29; 4	29; 4	−0.236_(138)_; 0.814; 0.044
%BF (%) ^b^	32; 8	34; 8	−1.394_(134)_; 0.166; 0.266
**Gender (n; %)**			**χ^2^_(df)_; *p*; φ**
Female	49; 33	32; 21	8.488_(1)_; 0.004; −0.239
Male	56; 38	12; 8	
**Education, class (n; %)**			**χ^2^_(df)_; *p*; φ**
<4 years	82; 55	29; 19	2.424_(1)_; 0.120; 0.128
≥4 years	23; 15	15; 10	
**BMI class (n; %)**			**χ^2^_(df)_; *p*; φ_c_**
Normal	17; 12	5; 3	0.756_(2)_; 0.679; 0.073
Overweight	47; 34	19; 14	
Obesity	35; 25	17; 12	
**MNA score (mean; SD)**			**t_(df)_; *p*; Cohen’s d**
MNA score (points)	27; 2	26; 3	2.731_(138)_; 0.007; 0.490
**Nutritional status (n; %)**			**χ^2^_(df)_; *p*; φ**
Risk of malnutrition	11; 8	8; 6	1.745_(1)_; 0.277; −0.112
Normal	88; 63	33; 24	
**Inflammatory indices (mean; SD)**			**Z_(U)_; *p*; *r*^£^**
hsCRP (mg/dL) ^¥ £^	1.56; 2.42	2.90; 2.27	−2.880_(1617.5)_; 0.004; 0.236

^a^*n* = 140 (iron sufficiency = 99 (70.71%), iron deficiency = 41 (29.29%)); ^b^
*n* = 136 (iron sufficiency = 97 (71.32%), iron deficiency = 39 (28.68%)). ^¥^ Variables not normally distributed. Data presented as median and interquartile range (median, IQR); ^£^ Mann–Whitney U test, results presented as Z_(U)_; *p*; r.

**Table 2 nutrients-12-03594-t002:** Hierarchical regression models for hematological dimensions predicting cognition, mood and physical functional ability based on hematological dimensions.

	Executive ^a^	Memory ^a^	MMSE ^a^	GDS ^b^	Functional-T ^c^	Functional-H ^c^
Storage (β; *p*)	0.112; 0.131	0.167; 0.037	0.051; 0.506	−0.144; 0.085	0.094; 0.248	0.002; 0.976
R^2^_adjusted_; F; *p*	0.377; 14.804; <0.0001	0.289; 9.679; <0.001	0.248; 8.971; <0.001	0.115; 4.773; <0.001	0.188; 7.281; <0.001	0.165; 6.375; <0.001
Transport (β; *p*)	0.014; 0.857	0.124; 0.134	0.096; 0.221	−0.176; 0.036	0.091; 0.283	−0.055; 0.521
R^2^_adjusted_; F; *p*	0.366; 14.178; <0.001	0.277; 9.166; <0.001	0.254; 9.218; <0.001	0.124; 5.115; <0.001	0.186; 7.235; <0.001	0.168; 6.477; <0.001
Transport S. (β; *p*)	0.042; 0.556	0.070; 0.383	0.038; 0.618	−0.181; 0.025	−0.018; 0.827	−0.039; 0.633
R^2^_adjusted_; F; *p*	0.367; 14.265; <0.001	0.268; 8.808; <0.001	0.247; 8.927; <0.001	0.128; 5.257; <0.001	0.180; 6.953; <0.001	0.166; 6.431; <0.001
Red cells C. (β; *p*)	0.094; 0.196	0.084; 0.300	0.051; 0.504	−0.078; 0.354	0.148; 0.069	−0.013; 0.879
R^2^_adjusted_; F; *p*	0.364; 14.040; <0.001	0.266; 8.731; <0.001	0.243; 8.758; <0.001	0.085; 3.690; 0.004	0.201; 7.827; <0.001	0.153; 5.905; <0.001
Erythropoiesis (β; *p*)	0.007; 0.930	0.227; 0.015	0.074; 0.412	−0.301; 0.001	0.197; 0.032	−0.012; 0.902
R^2^_adjusted_; F; *p*	0.355; 13.587; <0.001	0.295; 9.916; <0.001	0.244; 8.811; <0.001	0.144; 5.886; <0.001	0.209; 8.173; <0.001	0.153; 5.903; <0.001

^a^ Controlled for age, education, gender, high sensitivity C reactive protein (hsCRP) (log) and GDS; ^b^ controlled for age, education, gender and hsCRP (log); ^c^ controlled for age, gender, body mass index (BMI) and hsCRP (log).

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
