# Peer review of "Iron Status is Associated with Mood, Cognition, and Functional Ability in Older Adults: A Cross-Sectional Study"

_nutrients, 2020, doi:10.3390/nu12113594_

Round 1
Reviewer 1 Report
This is an interesting study on the association of iron status with mood, depression, functional ability and other related parameters.
There are some errors in the use of the English language through-out that will require careful editing, but overall the manuscript is well written and clear.
My only reservation and only reason for recommending a major revision is that one aspect of the authors' conclusions are not supported by their research. They state that "associations with iron status may be better recognized when iron levels are considered as continuum" and similarly in the manuscript's conclusion that this method is "more accurate and reliable."
This conclusion is not supported because the authors do not have a "gold standard" proving which of the two methods is actually valid. The gold standard by which ID/IDA can be diagnosed is bone marrow biopsy - and I am not suggesting that this test should have been carried out in these study subjects, rather that diagnosing true iron deficiency - especially iron deficiency without anaemia- is challenging. Unless the ferritin is below normal - and not an "adjusted below normal" but truly below normal for that laboratory - all the other variables are subject to variations not related to true iron deficiency (such as the diurnal variation in Fe levels) and confounding.
Nor do they have enough robust, interventional evidence agreeing with the continuum results vs the other results to justify their conclusion.
Below follow minor specific points:
Please check grammar in first two sentences of abstract
Please check grammar in first sentence introduction
In general I think "aged population" and "elderly" may be considered pejorative descriptions of older adults - please consider using other descriptors
Lines 45-46 - please check punctuation
Lines 50-51 - please clarify what kind of studies provided this evidence
Line 54 - please specify what kind of studies were analysed in the meta-analysis - interventional? epidemiological?
Line 112 - the ferritin level used here to define ID seems rather high compared with most lab values where I work - is this the standard for the authors' hospitals/clinics?
I find the tables challenging to read/interpret because of all the information presented - the authors are to be commended for providing this level of detail but I am wondering if they can present it in a less crowded table or tables?
line 240 - what are "almost hematological variables"?
Line 286 - what evidence do the authors have for this statement that "probably because ID was not sufficiently severe"?
Author Response
- This is an interesting study on the association of iron status with mood, depression, functional ability and other related parameters. There are some errors in the use of the English language through-out that will require careful editing, but overall the manuscript is well written and clear.
My only reservation and only reason for recommending a major revision is that one aspect of the authors' conclusions are not supported by their research. They state that "associations with iron status may be better recognized when iron levels are considered as continuum" and similarly in the manuscript's conclusion that this method is "more accurate and reliable." This conclusion is not supported because the authors do not have a "gold standard" proving which of the two methods is actually valid.
The gold standard by which ID/IDA can be diagnosed is bone marrow biopsy - and I am not suggesting that this test should have been carried out in these study subjects, rather that diagnosing true iron deficiency - especially iron deficiency without anaemia- is challenging. Unless the ferritin is below normal - and not an "adjusted below normal" but truly below normal for that laboratory - all the other variables are subject to variations not related to true iron deficiency (such as the diurnal variation in Fe levels) and confounding. Nor do they have enough robust, interventional evidence agreeing with the continuum results vs the other results to justify their conclusion.
Thank you for your positive comments on this manuscript, and for the careful and detailed evaluation of our manuscript. Your suggestions and concerns allowed a revision that will certainly be clearer and of interest to the readers. Below, please find our detailed responses to the points raised.
Major point
We understand your major concern and, in fact, an unfortunate typo mistake probably caused it. The ferritin levels considered for iron deficiency were <15 not 45 ng/mL. We are grateful that you noticed it.
Our intention when highlighting the usefulness of considering iron levels as a continuum, was not to criticize the need of cut-offs. These are very important. We have now rephrased the sentences where we refer to iron as a continuum.
In the abstract it now reads:
“These findings suggest that using iron as a continuous variable may be useful in finding associations with iron homeostasis, eventually missed when iron levels are considered within the usual classification groups.”
In the final paragraph of discussion, it now reads:
“In summary, the viewpoint that iron deficiency does not have consequences until the development of anemia is here challenged. Considering iron status as a continuum seems may provide relevant information to assess the consequences of low iron levels. In conclusion, low iron status in older individuals is associated with higher depressive mood, higher tiredness, and lower memory, which seem to be modulated by nutritional status. Further research is needed to replicate and confirm the present findings, including studies with longitudinal or interventional designs.”
Below follow minor specific points:
- Please check grammar in first two sentences of abstract
These sentences have been rephrased: “Several conditions are risk factors for iron deficiency (ID), some of which are highly prevalent in older individuals. Despite the amount of evidence pointing for a role of ID in cognition, mood and physical functional ability, the research addressing these associations in older individuals is still scarce.”
- Please check grammar in first sentence introduction
The sentence has been rephrased:“Imbalance in iron homeostasis, both excess and deficiency, are deleterious to human health and have been associated with medical conditions; these include neurodegenerative disorders (i.e. Parkinson and Alzheimer diseases), type II diabetes, and anemia.”
- In general, I think "aged population" and "elderly" may be considered pejorative descriptions of older adults - please consider using other descriptors
Elderly and aged population are terms widely used in scientific literature. However, we understand the reviewer’s concern and replaced those terms by others such as “older population”, “older individuals” or participants or “older adults”.
- Lines 45-46 - please check punctuation
Revised.
- Lines 50-51 - please clarify what kind of studies provided this evidence
We have now included the information on the type of studies that provide such evidence and the references after each one of the conditions.
“In adults, data from epidemiological and interventional studies indicate that ID can result or be associated with a wide range of adverse effects, including: fatigue [13, 14], reduced work performance, diminished exercise capacity [15], impaired thermoregulation [16], immune dysfunction [17], and neurocognitive impairment.”
- Line 54 - please specify what kind of studies were analyzed in the meta-analysis - interventional? epidemiological?
The nature of the studies - “epidemiological longitudinal studies” - has been added: “A systematic review and meta-analysis of epidemiological longitudinal studies [14] found an increased risk of incident dementia in anemic individuals; however, the type of anemia was not addressed.”
- Line 112 - the ferritin level used here to define ID seems rather high compared with most lab values where I work - is this the standard for the authors' hospitals/clinics?
We are very grateful that you identified this error, for which we apologize. 45 ng/mL should read 15 ng/mL. The appropriate correction was made in the text.
- I find the tables challenging to read/interpret because of all the information presented - the authors are to be commended for providing this level of detail but I am wondering if they can present it in a less crowded table or tables?
We agree on the importance that Tables convey their message, including all relevant information, and that the message is readable. We revised the Tables as an attempt to fulfill both criteria. We revised the number of significant digits as appropriate, which makes appearance “lighter” and have also included the p values on Table 2 (as requested by one of the referees).
- line 240 - what are "almost hematological variables"?
An “all” was missing from the sentence: “As expected, significant differences were observed for almost all hematologic variables and dimensions between groups of iron status (Table A4), with the iron sufficient group presenting higher values.”
- Line 286 - what evidence do the authors have for this statement that "probably because ID was not sufficiently severe"?
Our assumption is based on the observation that only 10 out of the 44 participants in the ID group presented ferritin levels below 15 ng/mL.
Reviewer 2 Report
General comments
Generally, the paper is well-written and logical.
The manusript describes a cross-sectional human study to look at the associations between iron status, nutrition status, mood, cognitive ability, and physical functional performance in community-dwelling middle-aged/older individuals, without dementia and/or pathology. Iron status was assessed using several blood measurements and categorised into storage, transport, etc.
Manuscript findings were that lower iron status is associated with depression, functional tiredness and poorer memory, the latter, moderated by nutritional status. Also, findings suggest that treating assessments of iron homeostasis as a continuum rather than categorial (ID or not ID) provides associations not observed with the latter.
Major Comments
There are interesting findings with respect to MNA scores and storage on memory (see also item 8 of discussion). However, while MNA is briefly said to assess protein-energy nutrition, the authors have not considered that the protein source may be meat which has a major source of iron – this needs to be discussed in more detail and with respect to the iron measures.
Also, some discussion is required regarding how systemic iron measurements can relate to the brain, e.g., brain iron accumulates with ageing, etc.
Minor Comments
Abstract
What aspect of nutrition, is nutritional status (line 20) referring to? Should iron status also be included here.
Introduction
1. Lines 36 and 56, define iron deficiency and iron deficiency anemia, as they are important for the article or ensure that this is clearly written in the methods/results.
2. Line 42, give example of morbidities and therapeutic drugs that are highly prevalent in older individuals that are possible cause of ID.
Materials and methods
1. Line 69, define N2 and N1?
2. Line 71, what is a ‘convenience’ sample? This sentence is unclear, ‘303 individuals were contacted until sample size reached’, please clarify.
3. Were ID and iron sufficient group balanced for gender, it doesn’t appear so.
4. How were cognitive impairment and dementia diagnosis determined for initial exclusion criteria.
5. Re section 2.4, please provide more info re the mini nutritional assessment, e.g., does it provide an assessment of dietary iron intake, or other metals/micronutrients, if so, please detail.
Results
Section 3.1
1. There is some confusion as there appears to be 149 participants but in the various tables, each measurement seems to have different samples sizes, please clarify/explain why the group sizes are different for different measurements and justify inclusion/exclusion of participants in methods and state clearly what the sample sizes are.
2. Re table 1, is the iron deficient group shorter than iron sufficient group? Looking at gender, the p-value is 0.004, suggesting that there is a lower percentage of males that are iron deficient, and would that explain the significant height differences?
Section 3.2
1. Please show the PCA plots.
2. Line 205-206, please explain what is meant by low communalities in the solution and excluded from what model?
3. Line 240, were significant differences observed for almost all hematologic variable? ‘all’ appears missing from the sentence.
Section 3.5
1. Please include p-values in tables 2 and A6, and other tables that do not include the p-value.
2. Also why was gender not controlled for in functional-T and functional-H? especially as all hematological dimensions predicted functional-T and there appeared a gender effect according to Table 1.
3. Please explain table A6, there seems to be a lot of significances, but authors say only memory and storage are significant.
Discussion
1. The authors say that using the usual cut offs for iron sufficiency and deficiency, there were no associations between iron status, cognition, mood and physical ability, but the ID group did have a high GDS than the iron sufficient group, please clarify.
2. On line 289, cohort subjects have been identified to be ID, ID+anemia, anemia and normal – this should be mentioned in results, are the Iron sufficient and ID individuals with anemia included in stats testing?
3. On line 297, was there no association with MMSE as it was excluded from the final model because of ‘low communalities’? For executive dimension, that was included in the final model and found not to be significantly associated with the erythropoiesis dimension?
4. Please explain why the iron storage dimension was also a predictor of memory (line 299).
5. Line 301, please state if the association between ID and MMSE is in aged subjects with no pathology.
6. The ID group showed significantly higher hsCRP (table 1), suggestive of increased inflammation, so the depression observed in the ID group could be due to inflammation (see 310-313, also point 9)?
7. Re line 321-322.
8. There are interesting findings with respect to MNA scores and storage on memory. Therefore, more info needs to be provided to say what MNA scores measure and mean to properly understand the results presented. Re lines 326 – 328, MNA test is stated to detect protein-energy malnutrition and not micronutrient deficiency. However, I would suggest with sufficient malnutrition, if the source of the protein is meat, then it is very likely that MNA will be related to iron intake? Further, in table 1, MNA score is lower in the ID group.
9. Line 341, the authors suggest while hsCRP are raised, they are not high enough to suggest an inflammatory state, please state the level at which hsCRP levels would be considered an inflammation state.
10. Line 353, states that this is the first study to examine iron status, in the absence of anemia, with cognition, etc, however, previously (line 289), some subjects were anemic, please clearly state if they were excluded.
11. Line 356, may be ‘convenience’ is meant to be ‘convenient’?
12. In the final paragraph, I suggest that ref 24 also challenges the viewpoint that iron deficiency does have consequences even when anemia is not present. The discussion should discuss in more detail about the manuscript findings with that in ref 24, in particular.

Author Response
Reviewer 2
- Generally, the paper is well-written and logical. The manusript describes a cross-sectional human study to look at the associations between iron status, nutrition status, mood, cognitive ability, and physical functional performance in community-dwelling middle-aged/older individuals, without dementia and/or pathology. Iron status was assessed using several blood measurements and categorised into storage, transport, etc.
Manuscript findings were that lower iron status is associated with depression, functional tiredness and poorer memory, the latter, moderated by nutritional status. Also, findings suggest that treating assessments of iron homeostasis as a continuum rather than categorial (ID or not ID) provides associations not observed with the latter.
Thank you for your positive comments on this manuscript, and for the careful and detailed evaluation of our manuscript. Your suggestions and concerns allowed a revision that will certainly be clearer and of interest to the readers. Below, please find our detailed responses to the points you raised.
Major Comments
- There are interesting findings with respect to MNA scores and storage on memory (see also item 8 of discussion). However, while MNA is briefly said to assess protein-energy nutrition, the authors have not considered that the protein source may be meat which has a major source of iron – this needs to be discussed in more detail and with respect to the iron measures.
We have now included additional information on the MNA scale, which does not allow to infer from the source of protein. “MNA is a validated questionnaire designed to provide a single and rapid measure of nutritional status in older individuals. It is composed by anthropometric measures, questions related to medication, mobility, autonomy of feeding, number of meals, food and fluid intake and self-perception on health and nutrition [34]. With respect to food intake, there are three questions (At least one serving of dairy products (milk, cheese, yoghurt) per day; Two or more servings of legumes or eggs per week and Meat, fish or poultry every day).”
We have also further discussed this aspect in the revised discussion section.
- Also, some discussion is required regarding how systemic iron measurements can relate to the brain, e.g., brain iron accumulates with ageing, etc.
We have now included some discussion on these aspects:
“During the aging process a certain degree of brain iron accumulation is observed and may be deleterious [54]. Insights from experiments on animal models of iron deficiency provide clues that possibly explain this finding; particularly in brain areas related to memory such as the hippocampus [55]. ID has been shown, in rats, to influence resting energy status, neurotransmission and myelination [56]. The precise molecular mechanisms mediating its effects, particularly during aging, remain to be identified and are likely to involve molecules such as hippocampal brain-derived neurotrophic factor [57].”
Minor Comments
Abstract
- What aspect of nutrition, is nutritional status (line 20) referring to? Should iron status also be included here.
The MNA scale provides general information on nutrition, without detailing on micronutrients intake, as specified above.
Introduction
- Lines 36 and 56, define iron deficiency and iron deficiency anemia, as they are important for the article or ensure that this is clearly written in the methods/results.
We have now revised, and iron deficiency anemia is defined as ID anemia.
- Line 42, give example of morbidities and therapeutic drugs that are highly prevalent in older individuals that are possible cause of ID.
The information requested was added: “On this, of relevant note, several morbidities (e.g.: gastric and duodenal ulcers, adenomatous polyps and erosive gastritis) and therapeutic drugs (e.g.: antacids, H2 antagonists, proton pump inhibitors, aspirin or nonsteroidal anti-inflammatory drug), that are prevalent in older individuals, are also possible causes of ID [10]”.
Materials and methods
- Line 69, define N2 and N1?
N1 was the expected sample size for ID group and N2 was the expected sample size for iron sufficiency group. We have now rephrased to make it clearer: “Since the first aim of this work is to explore differences in cognition, mood and physical functional ability between individuals with ID and iron sufficiency, sample size was calculated using GPower (v. 3.1.9.7) for the difference between two independent means considering a medium effect size (d = .5), power = .8, α = .05 and allocation ratio iron sufficiency group/ID group = .3. This estimation indicated a sample size of 142 individuals (iron sufficiency group = 109 and ID group = 33).”
- Line 71, what is a ‘convenience’ sample? This sentence is unclear, ‘303 individuals were contacted until sample size reached’, please clarify.
The sample was selected through a convenience sampling procedure, since the selection procedure did not consider the representativeness of the population. To make it clear we rephrased the sentence: “A convenience sample of 303 individuals was contacted and invited to participate from primary health care centers (Braga and Guimarães/Vizela) and internal medicine outpatient care (Hospital de Braga), until the calculated sample size was reached. We further included 20 more participants to account for exclusions and drop-outs.”
- Were ID and iron sufficient group balanced for gender, it doesn’t appear so.
You are right, they are not gender balanced. Given the selection strategy (convenience sample) no gender considerations were taken in participants recruitment. Gender was rather included and controlled for in the analyses.
- How were cognitive impairment and dementia diagnosis determined for initial exclusion criteria.
We have revised the sentence for clarification: “The assessment of exclusion criteria was initially based on self-report; the presence or absence of medical conditions was next confirmed from medical records.”
- Re section 2.4, please provide more info re the mini nutritional assessment, e.g., does it provide an assessment of dietary iron intake, or other metals/micronutrients, if so, please detail.
Please see response to the first major comment.
Results
Section 3.1
- There is some confusion as there appears to be 149 participants but in the various tables, each measurement seems to have different samples sizes, please clarify/explain why the group sizes are different for different measurements and justify inclusion/exclusion of participants in methods and state clearly what the sample sizes are.
We have now specified the reason for not having all variables for the 149 participants enrolled in the study: “In some measures, we were not able to collect data from participants, due to refusal of participants or due to contraindications, such as pacemaker in bioelectrical impedance analysis (BIA). Even with missing data for some measures we decided to use all available data from each participant. The sample size for each variable is presented as footnote in the tables.” (Section 2.1-Subjects).
- Re table 1, is the iron deficient group shorter than iron sufficient group? Looking at gender, the p-value is 0.004, suggesting that there is a lower percentage of males that are iron deficient, and would that explain the significant height differences?
Yes, you are right. We have now specified that possibility in the revised text: “Height was significantly higher in the iron sufficient which can be attributed to the differences in gender distribution since more male individuals were in this group.”
Section 3.2
- Please show the PCA plots.
Unfortunately, the software used (SPSS) does not provide the PCA plots.
- Line 205-206, please explain what is meant by low communalities in the solution and excluded from what model?
Communalities are the proportion of each variable's variance that can be explained by the principal components. In the case, we included the DS forward and MMSE total score in the PCA, but they presented low communalities and, therefore, were excluded from the component. Communality is a deciding factor to exclude variables from the principal component analysis. To make it clear we rephrased the corresponding sentence: “…executive dimension and (ii) memory dimension. The executive dimension (Cronbach’s alpha: 0.880) was composed of the Stroop (W, C and W&C) and DS (backward and total) parameters. The variables DS forward and MMSE total score were included in the initial analysis; however, since they provided low communalities, were later removed from the final model.”
- Line 240, were significant differences observed for almost all hematologic variable? ‘all’ appears missing from the sentence.
We corrected it accordingly.
Section 3.5
- Please include p-values in tables 2 and A6, and other tables that do not include the p-value.
Information included.
- Also why was gender not controlled for in functional-T and functional-H? especially as all hematological dimensions predicted functional-T and there appeared a gender effect according to Table 1.
You are right. We should have controlled for sex also for the functional variables, which we have now done in the revised version. The association remains significant for the erythropoiesis dimension.
- Please explain table A6, there seems to be a lot of significances, but authors say only memory and storage are significant.
We specifically addressed the significance of the moderation effect of MNA in the association of storage with Memory. That significant result is obtained from the interaction term of storage*MNA (see table A6). To allow a graphical visualization of the effect of MNA in the association of storage with Memory we created Figure 3. No other interaction term (hematological dimension*MNA) was a significant predictor of the dependent variables. Indeed, there is a lot of other significances presented in table A6; however, those significances concern to the predictive value of hematological variables on dependent variables, and those were already explored in the Table 2, and to the predictive value of MNA on dependent variables, that is not under the scope of our work and had been previously addressed in the scientific literature.
Discussion
- The authors say that using the usual cut offs for iron sufficiency and deficiency, there were no associations between iron status, cognition, mood and physical ability, but the ID group did have a high GDS than the iron sufficient group, please clarify.
On figure 2 it is stated that there is a significant difference in GDS scores between iron deficiency and iron sufficiency groups; yet, that result is from a crude analysis, that is also shown in table A1. We also tested the differences controlling for main confounding factors using ANCOVA and no differences in GDS scores were observed for those groups (table A3). That is mentioned on the last paragraph of section 3.2: “After controlling for principal confounding factors there were no differences between groups of iron status for mood, cognition and physical functionality (Table A3).”
- On line 289, cohort subjects have been identified to be ID, ID+anemia, anemia and normal – this should be mentioned in results, are the Iron sufficient and ID individuals with anemia included in stats testing?
All individuals were included in the analysis. We have now detailed the information on the number of subjects with anemia: “From the 149 participants, n=44 individuals (29.53% |ID=36; ID anemia=8) were classified as ID and n=105 individuals (70.47% | anemic=15; normal=90) as iron sufficient. No participant presented severe anemia (Hb < 9mg/dL).” (Section 3.1)
- On line 297, was there no association with MMSE as it was excluded from the final model because of ‘low communalities’? For executive dimension, that was included in the final model and found not to be significantly associated with the erythropoiesis dimension?
From a theoretical point of view, one would expect MMSE to be included in the principal component termed Executive function. Statistical criteria indicated that it should not be included (low communalities). For that reason, we made separate analyses for the component Executive dimension and for MMSE. In both cases, no associations were observed with the erythropoiesis dimension.
- Please explain why the iron storage dimension was also a predictor of memory (line 299).
Discussion on the association of iron and memory was added to the text
“Interestingly, the iron storage dimension was also a predictor of memory. ID is recognized to be deleterious for learning and memory, particularly when occurring in critical time windows of brain development [52]. These effects, which may remain into adulthood [55, 56], have been associated with abnormal hippocampal structure and plasticity [52].”
- Line 301, please state if the association between ID and MMSE is in aged subjects with no pathology.
From the reference we can conclude that the participants in the mentioned work were in general good health status since the exclusion criteria were:
- Metabolic and endocrinologic diseases that may cause cognitive dysfunction, including vitamin B12 deficiency, folate deficiency and hypothyroidism.
- Severe dementia (Stage 6 and 7 in Global Deterioration Scale).
- Patients to whom MMSE test could not be performed due to poor cooperation, because of visual problems, auditory problems, severe dementia, and decompansation in genaral situation.
- Patients who were previously diagnosed with iron deficiency, so were on iron treatment.
- Beta thalassemia trait and other hemoglobinopathies
- Myelodysplastic syndrome
- Transferrin saturation >50%
We added this information in the text: “Interestingly, others have described an association between ID and MMSE in “healthy” older adults [24].”
- The ID group showed significantly higher hsCRP (table 1), suggestive of increased inflammation, so the depression observed in the ID group could be due to inflammation (see 310-313, also point 9)?
Indeed, the depressive mood mentioned may be due to inflammation; however, we cannot say that participants presented an inflammatory profile because individuals with levels of hsCRP >10 mg/L were excluded. Furthermore, as previously mentioned, when we controlled for the main confounding factors (including hsCRP) no significant differences were observed in depressive mood between groups.
- Re line 321-322.
We apologize, from reading lines 321-322 we could not infer from the potential question of the reviewer.
- There are interesting findings with respect to MNA scores and storage on memory. Therefore, more info needs to be provided to say what MNA scores measure and mean to properly understand the results presented. Re lines 326 – 328, MNA test is stated to detect protein-energy malnutrition and not micronutrient deficiency. However, I would suggest with sufficient malnutrition, if the source of the protein is meat, then it is very likely that MNA will be related to iron intake? Further, in table 1, MNA score is lower in the ID group.
Please see the response to the major comment above.
- Line 341, the authors suggest while hsCRP are raised, they are not high enough to suggest an inflammatory state, please state the level at which hsCRP levels would be considered an inflammation state.
That information passed unnoticed from the exclusion criteria on section 2.1: “As shown in figure 1, n=162 participants accepted to participate in the study. From those, 11 subjects were further excluded due to: malnutrition (n=2, MNA score <17), chronic kidney disease (n =1; creatinine >2.5 mg/dL), high inflammatory status (n=7; hsCRP >10 mg/L) and iron overload (n=1; TF saturation >55%).”
- Line 353, states that this is the first study to examine iron status, in the absence of anemia, with cognition, etc, however, previously (line 289), some subjects were anemic, please clearly state if they were excluded.
We apologize for that mistake. The absence of anemia should not be there. We have corrected it accordingly in the text and as previously asked we state clearly in the text that participants with anemia were included in the analysis.
- Line 356, may be ‘convenience’ is meant to be ‘convenient’?
We have rephrased for clarification: “In addition, the use of a convenience sample recruitment strategy may be associated with selection bias; still, this does not hamper the associations observed.”
- In the final paragraph, I suggest that ref 24 also challenges the viewpoint that iron deficiency does have consequences even when anemia is not present. The discussion should discuss in more detail about the manuscript findings with that in ref 24, in particular.
We agree that the research from Yavuz et al. support our statement. We added that to the final paragraph and included other reference supporting that. The final paragraph now reads:
“In summary, the viewpoint that iron deficiency does not have consequences until the development of anemia is here challenged. Considering iron status as a continuum may provide relevant information to assess the consequences of low iron levels. This is accordance with the observations by Yavuz et al. [30], who showed that ID has negative consequences on cognitive function independently of the presence of anemia. Furthermore, ID seems to be deleterious even in the absence of erythropoietic effects [79]. In conclusion, low iron status in older individuals is associated with higher depressive mood, higher tiredness, and lower memory, which seem to be modulated by nutritional status. Further research is needed to replicate and confirm the present findings, including studies with longitudinal or interventional designs.”
Round 2
Reviewer 2 Report
The manuscript has been adequately revised according to comments/suggestions.